# Modeling cystic fibrosis patient prognosis: Nomograms to predict lung transplantation and survival prior to highly effective modular therapy

Annalisa V. Piccorelli [1]⊙*, Jerry A. Nick[2]⊙

1 Division of Biostatistics, Kaiser Permanente Washington Health Research Institute, Seattle, WA, United States of America, 2 Department of Medicine, National Jewish Health, Denver, CO, United States of America

⊙ These authors contributed equally to this work.
* Annie.V.Piccorelli@kp.org

## Abstract

### Background

The duration of time a person with cystic fibrosis (pwCF) spends on the lung transplant waitlist is dependent on waitlist and post-transplant survival probabilities and can extend up to 2 years. Understanding the characteristics involved with lung transplant and survival prognoses may help guide decision making by the patient, the referring CF Center and the transplant team.

### Methods

This study seeks to identify clinical predictors of lung transplant and survival of individuals with CF using 29,847 subjects from 2003–2014 entered in the Cystic Fibrosis Foundation Patient Registry (CFFPR).

### Results

Predictors significant ($p \leq 0.05$) in the final logistic regression model predicting probability of lung transplant/death were: $FEV_1$ (% predicted), BMI, age of diagnosis, age, number of pulmonary exacerbations, race, sex, CF-related diabetes (CFRD), corticosteroid use, infections with *B. cepacia*, *P. aeruginosa*, *S. aureus*, MRSA, pancreatic enzyme use, insurance status, and consecutive ibuprofen use for at least 4 years. The final Cox regression model predicting time to lung transplant identified these predictors as significant $FEV_1$ (% predicted), BMI, age of diagnosis, age, number of pulmonary exacerbations, race, sex, CF-related diabetes (CFRD), corticosteroid use, infections with *B. cepacia*, *P. aeruginosa*, *S. aureus*, MRSA, pancreatic enzyme use, and consecutive ibuprofen use for at least 4 years. The concordance indices were 0.89 and 0.92, respectively.

**Data Availability Statement:** The data underlying the results presented in the study are available from the Cystic Fibrosis National Patient Registry

([https://www.cff.org/medical-professionals/patient-registry](https://www.cff.org/medical-professionals/patient-registry)).

**Funding:** The author(s) received no specific funding for this work.

**Competing interests:** The authors have declared that no competing interests exist.

## Conclusions

The models are translated into nomograms to simplify investigation of how various characteristics relate to lung transplant and survival prognosis individuals with CF not receiving highly effective CFTR modulator therapy.

## Introduction

The lung function complications due to cystic fibrosis (CF) lead to death in 63% of persons with CF, and as a result, persons with CF (pwCF) are often listed for a lung transplant [1]. Treatment with the CF transmembrane-conductance regulator (CFTR) modulator ivacaftor was approved for the small cohort of patients with CFTR gating mutations in 2012 [2,3] which resulted in highly significant improvement in lung function, body mass index, quality of life, combined with a decreased rate of pulmonary exacerbations. CFTR modulators are small molecules that directly improve the activity and trafficking of the defective CFTR protein linked to specific CFTR mutations [4]. Combinations of modulators for additional CFTR genotypes were introduced in rapid succession, but with much more modest benefit. In late 2019 the triple combination elexacaftor, tezacaftor and ivacaftor (E/T/I) was approved in the US for pwCF homozygous [4] or heterozygous for F508del mutation [5]. E/T/I extended the benefits of "highly effective CFTR modulator therapy" (HEMT)to nearly90% of pwCF in the U.S.

Prior to introduction of HEMT the waiting period on the lung transplant list could last 2 or more years and 23% of CF patients may die while on the list, understanding when to list them is a priority [6,7] The Lung Allocation Score, which is used by the Organ Procurement and Transplantation Network to assure equitable organ allocation, takes survival probability while on the list and after transplantation into account, but is not tailored specifically to CF [8]. Following approval of E/T/I within the U.S. the 2020 CFFPR reported a notable reduction in the number of lung transplants performed [9].

People with CF are typically referred for lung transplant evaluation when their $FEV_1$ (forced expiratory volume in 1 second) falls below 30 (% predicted), with female patients and younger patients often considered earlier [10]. Studies have identified other patient characteristics involved in CF lung transplant and survival, although the set of the predictors chosen differs depending on the study [6,10–40]. Most of the studies indicate $FEV_1$% predicted (FEV1pp), age, number of pulmonary exacerbations, BMI, and sex are involved; however, the accuracy of these models often rely on many more.

Identifying the clinical characteristics involved in predicting when an individual will require a lung transplant may lead to a more personalized approach to transplant referral and listing. Additionally, these need to be communicated effectively from clinicians at CF Care Centers to pwCF and lung transplant centers [10]. The CFF consensus guidelines recommends "routine clinician-led efforts to discuss disease trajectory and treatment options, including lung transplantation" and "communication between the CF and lung transplant care teams at least every 6 months and with major clinical changes" [10].

The purpose of this study is to develop models to predict probability of lung transplant or death and time to lung transplant or death of CF patients in the United States prior to HEMT, and translate these into nomograms. Nomograms are meant to personalize estimations of survival and potentially ease communication between the CF clinician and patient and the lung transplant care team about the characteristics of the CF patient affecting both their probability of lung transplant or death and their probability of lung-transplant-free survival. Nkam et al.

(2017) developed a nomogram predicting a patient's 3-year risk of lung transplant or death given the patient's characteristics based on data from the French CF Registry [30]. The current study aims to demonstrate the validity of this method using data pre-HEMT, as a step towards a current prediction of differing time periods of risk based on patient-specific characteristics to provide a more detailed understanding of a specific patient's prognosis.

## Materials and methods

### Patients

This study was approved by the University of Wyoming Institutional Review Board. A subset of data from the CFF Patient Registry (CFFPR) was used in our study. The CFFPR collects data on CF patients who have been treated in CFF-accredited institutions and consented to be included in the CFFPR [41]. The CFFPR is estimated to include approximately 81–84% of persons with CF in the United States [41]. The data included 29,847 patients ages 6 to 40 years from Jan. 1, 2003 to Dec. 31. 2014. Age was defined as age at follow-up and determined using the most recent observed review year and the year of birth. These age limits were included in this study, because pulmonary function tests cannot be measured reliably until age 6 and estimates of survival after age 40 are subject to survivor bias and severe left truncation [42]. Specific definitions were used based on whether the patient experienced the event. For patients who had lung transplant or died, age was calculated as age at least one year prior to the event. This enabled earlier predicted time for listing for lung transplant with the understanding that patients can be on the waitlist for up to 2 years. The indicator of the lung transplant was created based on a variable defined as the year of the first lung transplant. Patients' lives after this first transplant and patients with a prior transplant were not considered. We assumed earlier prediction would be beneficial for patients. For patients censored due to the end of the study period or lost to follow-up, age was calculated as age in the most recently observed year. This was a retrospective study of deidentified data and the authors did not have access to the patient identification. The interval chosen pre-dates significant usage of HEMT in the US CF population.

### Predictors and outcome variables

Lung transplant or death, whichever occurred first, was the event of interest for both the logistic and Cox regression models, described in more detail below. Since patient death indicated the patient likely needed a lung transplant, we included death as alternative event of interest of lung transplant. For the Cox model, time to event was set up as the minimum of the age in years from year of birth to 1 year prior to lung transplant or death, with age in the most recently observed review year used for censored observations. The data cleaning and summary measures calculations were conducted with SAS 9.4.

The predictor variables attempted are shown on Table 1 and are based on prior studies [6,10–40]. The following demographic variables were considered: race, sex, F508del genotype, and age of diagnosis. Race was defined as whites v. other, because the majority (93.7%) were white. The following variables that were collected at encounter visits were also considered: weight, height, BMI, most recently documented age, best yearly FEV1% predicted (FEV1pp), best yearly forced vital capacity % predicted (FVCpp), *Pseudomonas aeruginosa* (*P. aeruginosa*) infection, *Staphylococcus aureus* (*S. aureus*), methicillin-resistant *Staphylococcus aureus* (MRSA) infection, *Burkholderia cepacia* (*B. cepacia*) infection, CF related diabetes (CFRD), pancreatic enzyme use, high dose ibuprofen use for at least 4 consecutive years (high ibuprofen), corticosteroid therapy, insurance status, and number of pulmonary exacerbations treated with IV antibiotics (NumPulmExacerbation). All of the variables other than best yearly

**Table 1. Summary of demographic, annualized, and best yearly pulmonary characteristics by event.**

| Characteristic | | Death/Transplant 5206 (17.4%) | Alive 24,641 (82.6%) |
|---|---|---|---|
| Demographic | | | |
| Race* | White | 4979 (95.6%) | 23,033 (93.5%) |
| Sex* | Female | 2740 (52.6%) | 11,565 (46.9%)) |
| F508del genotype[m]* | 1 | 2514 (48.3%) | 10,907 (44.3%) |
| | 2 | 1595 (30.6%) | 9288 (37.7%) |
| | 3 | 388 (7.5%) | 3595 (14.6%) |
| Age of diagnosis* | Mean (SD) | 2.3 (4.7) | 3.5 (6.6) |
| Annualized variables 1 year prior to event or at censoring | | | |
| *B. cepacia*[m]* | Positive | 62 (1.2%) | 385 (1.6%) |
| CF related diabetes[m]* | 1 | 1924 (37.0%) | 17,237 (70.0%) |
| | 2 | 375 (7.2%) | 2662 (10.8%) |
| | 3 | 2907 (55.8%) | 4742 (19.2%) |
| On corticosteroid therapy[m]* | Positive | 1541 (29.6%) | 13,489 (54.7%) |
| *P. aeruginosa*[m]* | Positive | 2252 (43.3%) | 12,025 (48.8%) |
| *S. aureus*[m]* | Positive | 1516 (29.1%) | 16,603 (67.4%) |
| MRSA[m]* | Positive | 951 (18.3%) | 6331 (25.7%) |
| On pancreatic enzymes[m]* | Positive | 4889 (93.9%) | 20,985 (85.2%) |
| Insurance[m]* | Federal | 2677 (51.4%) | 8936 (36.3%) |
| | Private | 2345 (45.0%) | 14,901 (60.5%) |
| | Other | 27 (0.5%) | 209 (0.8%) |
| | None | 43 (0.8%) | 328 (1.3%) |
| High Ibuprofen use for ≥4 consecutive years* | Positive | 23 (0.4%) | 482 (2.0%) |
| BMI[m]†* | Mean (SD) | 19.6 (3.5) | 20.9 (4.2) |
| Height[m]* | Mean (SD) | 162.4 (12.6) | 157.5 (18.8) |
| Weight* | Mean (SD) | 52.3 (13.5) | 54.0 (19.4) |
| Age* | Mean (SD) | 26.5 (7.7) | 20.6 (9.6) |
| NumPulmExacerbation‡* | Mean (SD) | 1.9 (2.0) | 0.7 (1.3) |
| Best yearly 1 year prior to event or at censoring | | | |
| FEV1pp[m]* | Mean (SD) | 49.4 (27.2) | 83.0 (25.6) |
| FVCpp[m]* | Mean (SD) | 61.0 (23.3) | 92.5 (21.2) |

[m] Some observations were missing.

*significant difference between groups (p-value<0.05), association between categorical predictor and response.

†calculated from height and weight.

‡number of pulmonary exacerbations treated with IV antibiotics.

FEV1pp and FVCpp were based on annualized observations. Annualized indicates the data were converted from encounter visits; for quantitative variables, the highest measurement from each quarter was taken and then was averaged to create on measure for the year. Annualized observations were selected over encounter observations because there were less missing observations in the annualized dataset. Best yearly FEV1pp and FVCpp were calculated from encounter visits. Because there were more missing BMI values than height and weight values, BMI was calculated based on the formula: BMI = weight/height$^2$. Annualized and best yearly observations were taken 1 year prior to the death/lung transplant (whichever was first) or at time of censoring.

Correlation between predictors was assessed as follows: Pearson correlation between continuous predictors, polychoric correlation between categorical predictors with 2 or more

categories, and tetrachoric correlation between binary categorical predictors. In order to assess correlation between the continuous predictors and the categorical predictors, indicator variables for each category of the categorical predictors were created, and Pearson correlation was used to assess the correlation between each continuous predictor and each indicator-variable-form of the categorical variables. Strongly correlated predictors with |correlation|≥ 0.5, e.g. FEV1pp and FVCpp, were not fit in the same model to avoid multicollinearity.

The % missingness observed for these variables was as follows: F508del genotype (5.2%), *B. cepacia* (11.5%), corticosteroid therapy (13.0%), *P. aeruginosa* (11.5%), *S. aureus* (11.5%), MRSA (11.5%), pancreatic enzyme usage (1.5%), insurance (1.3%), FEV1pp (5.4%), FVCpp (5.3%), height (0.5%), weight (0.8%), age of diagnosis (0.1%), and BMI (1.4%). Missing observations were imputed using SAS PROC MI. Twenty-five datasets were imputed. The standard errors and means of each imputed dataset were compared and the differences among them was minimal, ranging from 0.000464 to 0.0000000000 and from 0.099111 to 0.000000000 for the means. Since the differences among the datasets, the imputed dataset that resulted in the minimum standard error for the majority of the quantitative variables was used for model development.

## Model development

The data was analyzed using the software R 4.0.3. Multiple logistic regression modeling was used to predict lung transplant or death and predict the probability of lung transplant or death. Cox regression models were used to identify significant predictors of time to lung transplant or death and probability of time to lung transplant or death. Full logistic and Cox regression models were fit including all independent variables of interest. Significance of predictors ($p \leq 0.05$), Akaike information criterion (AIC), Bayesian Information Criteria (BIC), and concordance index were used to select the best subset of predictors for the final models. The Holm method was used to correct p-values for multiple testing [43]. The Cox proportional hazards assumption was checked using Schoenfeld residuals. For all of the predictors, the Schoenfeld residuals randomly fluctuated around 0 demonstrating the proportional hazards assumption was held for each of the predictors. Final models were used to generate nomograms of the probability of lung transplant or death and time to lung transplant-free survival.

The models were constructed using a training dataset consisting of a randomly selected 90% of the observations in the dataset. Model classification to the correct observed event (lung transplant/death or no lung transplant/death) outcome was assessed using the testing dataset, the remaining 10% of the dataset.

## Model validation

The validation of the models was measured using concordance index, leave one out cross-validation, and bootstrapped resampling. Concordance index measuresthe proportion of the predictions of the models to predict lung transplant/death in patients that experienced lung transplant/death and was used to assess the fit of both the logistic and Cox regression models. A calibration plot was used to assess the goodness of the fit of the logistic regression model; specifically, the observed values for the testing dataset were plotted against the predicted values from the logistic regession model. Leave one out cross-validation was used to determine the estimated prediction error of the logistic regression model using R function cv.glm with the number of resamples equal to the sample size of 29,847. Since there is not an R function for leave one out cross-validation with Cox regression modeling, bootstrapped resampling with 1000 repetitions was used to validate the Cox regression model with R function validate.

# Results

## Patient characteristics

The subset of the CFFPR eligible for this study was 29,847 CF patients. Descriptive statistics of the study sample are shown on Table 1. Death or lung transplant was observed in 5206 (17.4%) patients with the remaining 24,641 patients (82.6%) censored in 2014 or if missing in 2014, the time of most current review prior to 2014. The median time of follow-up was 8 years. Significant differences between the death/transplant and the alive groups were observed in the following characteristics: age of diagnosis, BMI, height, weight, age, FEV1pp, FVCpp, and number of pulmonary exacerbations treated with IV antibiotics (NumPulmExacerbation). Significant associations were observed between vital status and the following categorical variables: race, sex, F508del genotype, CFRD, corticosteroid therapy, *B. cepacia*, *P. aeruginosa*, *S. aureus*, MRSA, pancreatic enzymes therapy, insurance status, and high dose ibuprofen use for at least 4 consecutive years.

## Multiple logistic regression model

The following characteristics were significant predictors of probability of lung transplant/death in the final logistic regression model (Table 2 and Fig 1): FEV1pp, BMI, age of diagnosis, age, NumPulmExacerbation, race, sex, CFRD, corticosteroid therapy, *B. cepacia*, *P. aeruginosa*, *S. aureus*, MRSA, pancreatic enzyme usage, insurance status, and consecutive high dose ibuprofen use for at least 4 years.

**Table 2. Logistic regression results for probability of lung transplant/death using complete dataset.**

|  | Estimated Odds Ratio | 95% Confidence Interval |
|---|---|---|
| **Sex (male)** | 0.77 | (0.72, 0.84) |
| **FEV1pp** | 0.96 | (0.961, 0.964) |
| **BMI** | 0.88 | (0.87, 0.89) |
| **NumPulmExacerbation‡** | 1.26 | (1.23, 1.29) |
| **Age** | 1.07 | (1.06, 1.07) |
| **Age of diagnosis** | 0.97 | (0.96, 0.98) |
| **Race (white)** | 1.53 | (1.29, 1.83) |
| ***B. cepacia*** | 0.41 | (0.30, 0.55) |
| ***P. aeruginosa*** | 0.39 | (0.36, 0.43) |
| ***S. aureus*** | 1.39 | (1.26, 1.53) |
| **MRSA** | 0.57 | (0.52, 0.63) |
| **CFRD2\*** | 0.81 | (0.71, 0.93) |
| **CFRD3\*** | 2.47 | (2.27, 2.68) |
| **Ibuprofen use for ≥ 4 consecutive years** | 0.25 | (0.15, 0.40) |
| **Insurance_none†** | 0.54 | (0.37, 0.78) |
| **Insurance_other†** | 0.53 | (0.33, 0.84) |
| **Insurance_private†** | 0.69 | (0.64, 0.75) |
| **Enzymes** | 1.84 | (1.54, 2.20) |
| **Corticosteroids** | 0.40 | (0.37, 0.43) |

‡number of pulmonary exacerbations treated with IV antibiotics.

*CFRD categories are indicator variables for CFRD2 = Impaired Glucose Tolerance, and CFRD3 = CFRD with or without fasting hyperglycemia with the reference category CFRD1 = Normal Glucose Metabolism.

†The reference category for insurance status was federal.

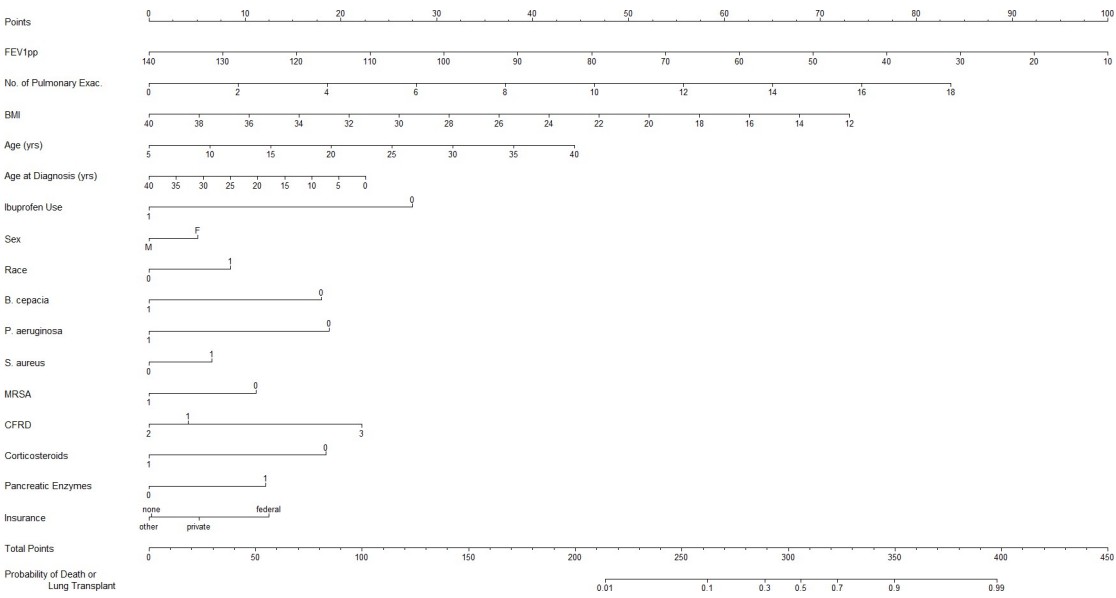

**Fig 1. Nomogram for probability of lung transplant or death.** For each characteristic, find the number of points by drawing a vertical line to the Points scale. Total the points for all the characteristics and draw a vertical line from the Total Points scale to get the probability of lung transplant or death. Notes about variables: No. of Pulmonary Exac: Number of pulmonary exacerbations treated with IV antibiotics in most recent year, Race: 0 = non-white, 1 = white; Sex: M = Male, F = Female; CFRD: 1 = Normal Glucose Metabolism, 2 = Impaired Glucose Tolerance, 3 = CFRD with or without fasting hyperglycemia; for variables, *B. cepacia*, *P. aeruginosa*, *S. aureus*, and MRSA: 0 = negative and 1 = positive; for variables, Corticosteroid Therapy (Corticosteroids) and Pancreatic enzyme usage (Pancreatic Enzymes): 0 = No, 1 = Yes; Insurance: F = Federal, P = Private, O = Other, and N = None; Ibuprofen use: Use of ibuprofen use for at least 4 consecutive years.

### Cox multiple regression model

The final Cox regression model of time to lung transplant/death (time in the form of age in years from birth) identified the following significant predictors as: FEV1pp, BMI, age of diagnosis, age, NumPulmExacerbation, race, sex, CFRD, corticosteroid therapy, *B. cepacia*, *P. aeruginosa*, *S. aureus*, MRSA, pancreatic enzyme use, and consecutive high dose ibuprofen use for at least 4 years (Table 3 and Fig 2).

### Model validation

With the calibration plot, the pairs of observed and predicted values of the outcome variable, lung transplant/death, closely on the 45˚ line with intercept at 0. This indicated agreement between the observed and predicted values and demonstrated the multiple logistic regession model provided a good fit. This was further supported by a concordance index of 0.89 and an estimated prediction error of 0.09 from leave one out cross-validation for the multiple logistic regression model. The Cox multiple regression model performed similarly as well with a concordance index of 0.92 based on the bootstrap resampling validation.

### Discussion

This study developed models that were able to accurately predict probability of the lung transplant/death and time to lung transplant/death. The logistic and Cox regression models were internally validated with accuracy of prediction at 89% and 92%, respectively. The logistic regression model predicting probability of lung transplant/death identified FEV1pp, BMI, age of diagnosis, age, NumPulmExacerbation, race, sex, CFRD, corticosteroid therapy, *B. cepacia*,

**Table 3. Cox regression results for time to lung transplant/death using the complete dataset.** Note time to age to lung transplant/death is measured as age from birth to lung transplant or death, whichever comes first, or age until censored for those that did not die or did not have lung transplant.

| | Estimated Hazard Ratio | 95% Confidence Interval |
|---|---|---|
| **Sex (male)** | 0.81 | (0.77, 0.86) |
| **FEV1pp** | 0.977 | (0.976, 0.979) |
| **BMI** | 0.89 | (0.88, 0.90) |
| **NumPulmExacerbation‡** | 1.09 | (1.07, 1.11) |
| **Age** | 0.60 | (0.59, 0.61) |
| **Age of diagnosis** | 0.98 | (0.97, 0.98) |
| **Race (white)** | 1.30 | (1.13, 1.48) |
| ***B. cepacia*** | 0.49 | (0.39, 0.62) |
| ***P. aeruginosa*** | 0.44 | (0.41, 0.46) |
| ***S. aureus*** | 1.51 | (1.41, 1.61) |
| **MRSA** | 0.57 | (0.53, 0.62) |
| **CFRD2** | 0.84 | (0.75, 0.94) |
| **CFRD3** | 1.98 | (1.86, 2.10) |
| **Ibuprofen use for $\geq$ 4 consecutive years** | 0.30 | (0.20, 0.46) |
| **Enzymes** | 1.80 | (1.55, 2.09) |
| **Corticosteroids** | 0.51 | (0.48, 0.54) |

‡number of pulmonary exacerbations treated with IV antibiotics.

**CFRD categories are indicator variables for CFRD2 = Impaired Glucose Tolerance, and CFRD3 = CFRD with or without fasting hyperglycemia with the reference category CFRD1 = Normal Glucose Metabolism.

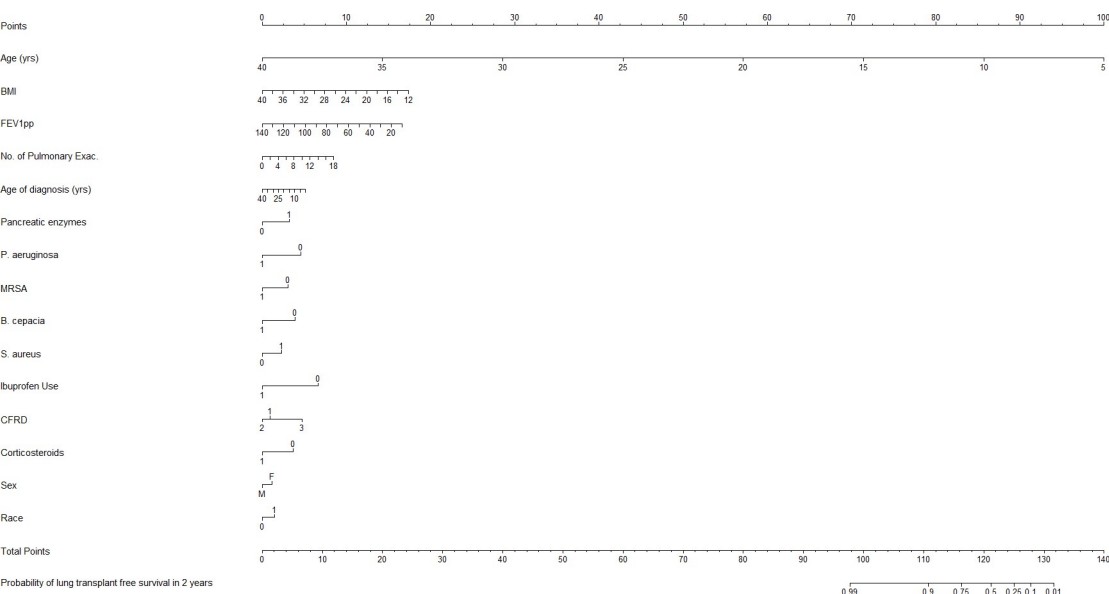

**Fig 2. Nomogram for probability of lung transplant or death free survival for 2 years.** Notes about variables: No. of Pulmonary Exac: Number of pulmonary exacerbations treated with IV antibiotics in most recent year, Race: 0 = non-white, 1 = white; Sex: M = Male, F = Female; CFRD: 1 = Normal Glucose Metabolism, 2 = Impaired Glucose Tolerance, 3 = CFRD with or without fasting hyperglycemia; for variables, *B. cepacia*, *P. aeruginosa*, *S. aureus*, and MRSA: 0 = negative and 1 = positive; for variables, Corticosteroid Therapy (Corticosteroids) and Pancreatic enzyme usage (Pancreatic Enzymes): 0 = No, 1 = Yes; Insurance: F = Federal, P = Private, O = Other, and N = None; Ibuprofen use: Use of ibuprofen use for at least 4 consecutive years.

*P. aeruginosa*, *S. aureus*, MRSA, pancreatic enzyme usage, insurance status, and consecutive high dose ibuprofen use for at least 4 years. Similarly the following characteristics were identified by the Cox regression modeling time to lung transplant/death: FEV1pp, BMI, age of diagnosis, age, NumPulmExacerbation, race, sex, CFRD, corticosteroid therapy, *B. cepacia*, *P. aeruginosa*, *S. aureus*, MRSA, pancreatic enzyme use, and consecutive high dose ibuprofen use for at least 4 years. Significant differences between and associations with vital status were observed in all the characteristics identified by the logistic and Cox regression models.

The logistic and Cox regression models were translated into nomograms. Nomograms are intended to ease the communication between clinicians and people with a disease. The characteristics included in the nomograms are measured at most encounter visits (see Methods section for more detail), thus, the CF clinician could calculate the CF patient's probability of lung transplant and probability of lung transplant-survival in 2 years and 5 years. This may enable the CF clinician and the individual with CF to stay on track with a timely lung transplant referral, to avoid complications and potential barriers to listing associated with late referrals [10]. This is intended help to address mortality prior to listing, which is reported as high as 50% in a recent study from France [44]. The additional benefit of the modeling time to lung-transplant free survival with a Cox regression is the ability to predict probability of lung transplant/death in different time-points allowing investigators to choose time-points that are meaningful to a specific person with CF.

Nkam et al. (2017) also developed a nomogram to predict probability of lung transplant. Their model predicted probability of lung transplant in 3 years based on a smaller sample (n = 2096) of the French CF Registry with only 3 years of data (2010–13) [30]. Their model included the following categorical predictors: *B. cepacia*, hospitalization (yes/no), oral corticosteroids, long-term oxygen therapy, and non-invasive ventilation, and categorical versions of FEV1pp ($\geq$ 60, [30–60], <30) and BMI ($\geq$ 18.5, [16–18.5], <16). Although the current study's models are more complex in terms are the number of predictors, they can be more specific to each patient given the models use non-categorical versions of the quantitative predictors and can predict probability of lung transplant-free survival at different time-points.

This study was limited by the variables available in the CFFPR. Characteristics suggested to exacerbate lung function and affect survival by other papers, such as infection with *Stenotrophomonas maltophilia*, 6 minute walking test, hypercarbia, and hypoxemia, were not available in the CFFPR [10,45–47]. In addition, various predictors in this nomogram may be specific to the United States, including types of insurance and other aspects relating to delivery of care that may be less obvious.

Although nomograms allow for easy communication, there are limitations involved with applying them to predicting probability of and timing of lung transplant/death. Nomograms do not accommodate time-varying covariates, which could have been useful with conditions that change prognosis after they are present, e.g., *B. cepacia*. Given the necessity of nomograms to allow for clear, easy-to-implement-and-interpret, although other models suggested slope of FEV1 and interactions between pulmonary exacerbations and FEV1pp as predictors, adding these into the model would have made the nomogram more complex to interpret [25,38]. Finally, the current nomograms were developed using data that generally predated widespread use of highly effective CFTR modulator therapy (HEMT) in the United States CF population. CFTR modulators improve outcomes for a majority of individuals with CF [3–5] and may deliver a sustained benefit over time [48]. Clinical trials excluded patients with advanced lung disease, thus there is little available data to guide the degree of disease modification in the period included within this study. It is now apparent that most individuals receiving HEMT will experience slower progression towards transplant or death in this cohort, but current CFF-sponsored guidelines for lung transplant referral suggest that transplant referral not be

delayed based on use of CFTR modulators [3–5,48]. Going forward, these findings can serve as a comparison for future cohorts who are on HEMT.

We have developed and internally validated nomograms to predict probability of lung transplant/death and probability of lung transplant-free survival in 2 year and 5 years. The nomograms are user-friendly and will facilitate further investigation into need for transplant and survival in people with CF with advanced lung disease.

## Acknowledgments

The authors would like to thank the Cystic Fibrosis Foundation for the use of the CF Foundation Patient Registry data to conduct this study. Additionally, we would like to thank the patients, care providers, and clinic coordinators at CF Centers throughout the United States for their contributions to the CF Foundation Patient Registry.

## Author Contributions

**Conceptualization:** Annalisa V. Piccorelli, Jerry A. Nick.

**Formal analysis:** Annalisa V. Piccorelli.

**Investigation:** Annalisa V. Piccorelli.

**Methodology:** Annalisa V. Piccorelli.

**Project administration:** Annalisa V. Piccorelli.

**Software:** Annalisa V. Piccorelli.

**Supervision:** Annalisa V. Piccorelli.

**Validation:** Annalisa V. Piccorelli.

**Visualization:** Annalisa V. Piccorelli.

**Writing – original draft:** Annalisa V. Piccorelli.

**Writing – review & editing:** Annalisa V. Piccorelli, Jerry A. Nick.

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
