## [Decision Letter · Decision Letter 0]

10 Jan 2024

PONE-D-23-30531MODELING CYSTIC FIBROSIS PATIENT PROGNOSIS: NOMOGRAMS TO PREDICT LUNG TRANSPLANTATION AND SURVIVAL PRIOR TO HIGHLY EFFECTIVE MODULATOR THERAPYPLOS ONE

Dear Dr. Piccorelli,

Thank you for submitting your manuscript to PLOS ONE. After careful consideration, we feel that it has merit but does not fully meet PLOS ONE’s publication criteria as it currently stands. Therefore, we invite you to submit a revised version of the manuscript that addresses the points raised during the review process.

We look forward to receiving your revised manuscript.

Kind regards,

Emrah Gecili, PhD

Academic Editor

PLOS ONE

Culture independent markers of nontuberculous mycobacterial (NTM) lung infection and disease in the cystic fibrosis airway - https://doi.org/10.1016/j.tube.2022.102276

In your revision ensure you cite all your sources (including your own works), and quote or rephrase any duplicated text outside the methods section. Further consideration is dependent on these concerns being addressed.

5. Please upload a new copy of Figures 1 and 2 as the detail were not clear. Please follow the link for more information: https://blogs.plos.org/plos/2019/06/looking-good-tips-for-creating-your-plos-figures-graphics/" https://blogs.plos.org/plos/2019/06/looking-good-tips-for-creating-your-plos-figures-graphics/

Reviewers' comments:

Reviewer's Responses to Questions

**Comments to the Author**

1. Is the manuscript technically sound, and do the data support the conclusions?

Reviewer #1: Partly

Reviewer #2: No

2. Has the statistical analysis been performed appropriately and rigorously? 

Reviewer #1: No

Reviewer #2: No

3. Have the authors made all data underlying the findings in their manuscript fully available?

Reviewer #1: No

Reviewer #2: Yes

4. Is the manuscript presented in an intelligible fashion and written in standard English?

Reviewer #1: Yes

Reviewer #2: Yes

5. Review Comments to the Author

Reviewer #1: Dear authors,

1. the subset of data you chosen include people with CF until their 40, limited to 2014. Age is "age at follow-up" or "age at registry entry"?

2. Despite the choice of limiting age at 40 could be likely based on the age at death extrapolated in the paper of Schluchter et al, censoring at 2014 can be seen as a majior limitation, consdidering the rapid changing clinical landscape of the disease, regardless HEMT. Also, many trials on HEMT were already ongoing during 2014. How these predictions could be useful now?

3. Considering the specific care of the CF disease, even back to 2014, lung transplant should be conceived as a competitive event of death;

4. With regard to missingness, it is not clear why you worked only on the imputed dataset with lower standard error and not on the combined imputed datasets (how many, for instance?);

5. By consequence, the validation as well should have be done on all imputed datasets.

6. Findings from the fitted model are however limited to an US audience only (e.g., variabile insurance not available in Europe)

7. Which time scale for the Cox Model?

Reviewer #2: This manuscript proposes a prediction model for lung transplantation and survival using standard logistic and Cox regression approaches. The authors utilized Cystic Fibrosis Foundation Patient Registry (CFFPR), which offers gigantic sample size with a small amount of missing data.

Major comments:

1. My major concern is the non-comparative approaches. Why authors did not try machine learning approaches such as lasso/random forest using rigorous cross-validations instead of a leave-one-out approach. Given this huge sample size, this is feasible. Here authors present prediction models, so I believe that it is essential to focus on prediction performances such as AUROC (sensitivity and specificity) rather than model goodness of fits. Correlations among predictors also need to be reported.

2. Again, with this huge sample sizes, the p-value <0.05 is not clinically informative. The choice of predictors needs to be guided by clinical interpretability. Any discussion as to the clinically meaningful associations in the prediction model?

3. Table 3, why there are so many predictors with p of 0.0016? Were these from copy and paste errors? Instead of presenting raw-estimates and SEs, it would be better if authors use HR and its 95% CI. Also this applies to Table 2 where authors can report ORs and their CIs. Please do not include intercepts in each Table.

4. About missing data, the authors used the multiple imputation. What if authors use the completely observed data only, will the result be identical?

5. As author mentioned, this cohort includes more whites than national-average. Any future accommodation for correcting this racial imbalance?

6. When reporting survival endpoint, it would be nicer if authors clarify the median follow-up time, frequency of censoring.

6. PLOS authors have the option to publish the peer review history of their article (what does this mean?). If published, this will include your full peer review and any attached files.

Reviewer #1: No

Reviewer #2: No

---

## [Author Response · Author response to Decision Letter 0]

20 Apr 2024

Culture independent markers of nontuberculous mycobacterial (NTM) lung infection and disease in the cystic fibrosis airway - https://doi.org/10.1016/j.tube.2022.102276

Thank you for identifying this overlapping text. We have extensively rewritten this introductory paragraph to insure it is completely original, and to improve the clarity of the text.

In your revision ensure you cite all your sources (including your own works), and quote or rephrase any duplicated text outside the methods section. Further consideration is dependent on these concerns being addressed.

Cystic Fibrosis Foundation Patient Registry (CFFPR) data were used for this study. The Cystic Fibrosis Foundation provides CFFPR data upon request, thus, it is not generally publicly available. The public can apply for CFFPR via this website: https://www.cff.org/researchers/patient-registry-data-requests and so others can access the data in the same manner we, the authors, obtained them. This is how we were approved to use the specific data used in the study, which was provided by the CFF directly. 

5. Please upload a new copy of Figures 1 and 2 as the detail were not clear. Please follow the link for more information: https://blogs.plos.org/plos/2019/06/looking-good-tips-for-creating-your-plos-figures-graphics/" https://blogs.plos.org/plos/2019/06/looking-good-tips-for-creating-your-plos-figures-graphics/

Reviewers' comments:

Reviewer's Responses to Questions

Comments to the Author

1. Is the manuscript technically sound, and do the data support the conclusions?

Reviewer #1: Partly

Reviewer #2: No

2. Has the statistical analysis been performed appropriately and rigorously? 

Reviewer #1: No

Reviewer #2: No

3. Have the authors made all data underlying the findings in their manuscript fully available?

Reviewer #1: No

Reviewer #2: Yes

4. Is the manuscript presented in an intelligible fashion and written in standard English?

Reviewer #1: Yes

Reviewer #2: Yes

5. Review Comments to the Author

Reviewer #1: Dear authors,

1. the subset of data you chosen include people with CF until their 40, limited to 2014. Age is "age at follow-up" or "age at registry entry"? 

We agree this information is important and we have made edits to further describe age in the Patients section of the Materials and Methods section.

Age is defined as age at follow-up. Age at follow-up is calculated as the difference between review year and date of birth year. Review year is the year when the clinical visit(s) occurred, so age is their age in that year. For patients that died or had a lung transplant, we calculated age at least one year prior to the event. We did this to enable additional time prior to the event of interest. Additionally, we assumed if a patient had a lung transplant or died, it would be beneficial for them to be listed for lung transplant earlier rather than later. For patients that were censored due to the end of the study period or lost to follow-up, we calculated their age as age in the most recently observed year. 

2. Despite the choice of limiting age at 40 could be likely based on the age at death extrapolated in the paper of Schluchter et al, censoring at 2014 can be seen as a major limitation, considering the rapid changing clinical landscape of the disease, regardless HEMT. Also, many trials on HEMT were already ongoing during 2014. How these predictions could be useful now?

We agree that the CF treatment landscape has evolved rapidly over the past two decades. In 2014 approximately 6% of the U.S. CF population had access to HEMT. A number of trials were ongoing for different combinations of modulators; however, these formulations are now recognized as providing very small incremental benefit and are rarely used today. It wasn’t until mid-2020 (following approval of E/T/I in November 2019) that the majority of pwCF had access to HEMT. As noted in the last sentence of the Introduction, “The current study aims to demonstrate the validity of this method using data pre-HEMT, as a step towards a current prediction of differing time periods of risk based on patient-specific characteristics to provide a more detailed understanding of a specific patient’s prognosis.” In the discussion, we more thoroughly explain our reasoning for using this data specifically: “Finally, the current nomograms were developed using data that generally predated widespread use of highly effective CFTR modulator therapy (HEMT) in the United States CF population. CFTR modulators improve outcomes for a majority of individuals with CF [2, 3, 5] and may deliver a sustained benefit over time [47]. Clinical trials excluded patients with advanced lung disease, thus there is little available data to guide the degree of disease modification in the period included within this study. It is now apparent that most individuals receiving HEMT will experience slower progression towards transplant or death in this cohort, but current CFF-sponsored guidelines for lung transplant referral suggest that transplant referral not be delayed based on use of CFTR modulators [2, 3, 5, 47]. Going forward, these findings can serve as a comparison for future cohorts who are on HEMT.” 

We acknowledge that these nomograms are not intended for clinical use. Extensive external validation would be required before introducing any tool into clinical care, and given the magnitude of the benefit from HEMT, it is certain that revisions are required. We do feel strongly that this method to determine probability and time to lung transplant or death is of use to build upon for the current CF population, and to model the need for transplant relative to historic trends and for those that do not benefit from HEMT. To the extent that a small number of pwCF receiving HEMT were included in this analysis only serves to make the analysis more relevant to the current population. We have reviewed the Discussion section and made edits to reinforce this point.

3. Considering the specific care of the CF disease, even back to 2014, lung transplant should be conceived as a competitive event of death.

We agree that lung transplant is a competitive event of death. That is why we defined the event as lung transplant or death, whichever came first. Reason for death was not well documented in the dataset and since lung disease is the cause of the death for the majority of CF patients, we assumed that if a CF patient died that indicated the patient needed a lung transplant .” (https://www.cdc.gov/scienceambassador/documents/cystic-fibrosis-fact-sheet.pdf). We added to the Methods section to further explain this.

4. With regard to missingness, it is not clear why you worked only on the imputed dataset with lower standard error and not on the combined imputed datasets (how many, for instance?);

We compared 25 different imputed datasets and the greatest differences among the standard errors were 0.000465 for FEV1 % predicted and 0.000000202 for BMI and the same for 10 decimal places for the other quantitative predictors, so although we chose the smallest standard error, the values were very close to each other or nearly identical. Similarly, the greatest differences among the means were 0.099111 for FEV1 % predicted, 0.008916 for BMI, 0.001396 for age of diagnosis, 0.000202 for minimum age of death/transplant and were identical for the other quantitative predictors. Thus, although we chose the smallest standard error, the standard error and mean values were very close to each other or nearly identical. We have incorporated details relating to the number of imputation conducted and our imputed dataset selection to the Methods section. We agree this is important information to include—thank you for bringing it to our attention.

5. By consequence, the validation as well should have be done on all imputed datasets.

As mentioned above in response to question 4, the differences between the standard errors and between the means of the quantitative predictors among the imputations were very small, so we chose not to repeat the analysis on all of the imputed datasets.

We chose to use leave one out cross-validation (LOOCV) to carry out internal validation of our logistic regression model and “bootstrapped resampling with 1000 repetitions was used to validate the Cox regression model with R function validate” (Model Validation section of the Methods section). Additionally, we built the models with a training dataset with a randomly selected dataset consisting of 90% of the observations and assessed the model accuracy with the remaining 10% of observations. The validation process is described in more detail in response to question 1 from reviewer 2 below and we have edited Methods section to explain this in more detail. 

6. Findings from the fitted model are however limited to an US audience only (e.g., variabile insurance not available in Europe)

Lung transplant allocation is decided by lung allocation score (LAS) for 60% of the world; however, countries have country/region-specific allocation systems (Gottlieb, 2017). Thus, we felt it was important to address this issue in a country-specific fashion. Ramos et al. (2017) found Medicaid insurance had an estimated hazard ratio of 1.67 risk of death in a univariate model and an estimated hazard ratio of 1.16 risk of death in a multivariate model. Ramos et al. (2017) found this using a retrospective cohort of the Cystic Fibrosis Foundation Patient Registry as we did with our study, so we felt it was important to assess whether the same predictors could be predictive of lung transplant/death. DuBay et al. (2017) also assessed the “frequency and variation in Medicaid transplantation, and post-transplant survival in Medicaid patients” and determined “Medicaid organ transplant beneficiaries had significantly lower survival compared to Privately insured beneficiaries;” this further supports our decision to assess the impact of insurance type on probability and time to lung transplant/death.

We agree that these results are based on the population of CF patients in the United States who are attending CF Care Centers, as the model was created using data was derived from the US Cystic Fibrosis Foundation Patient Registry. There are any number of differences in health care delivery and accessibility between various countries that could impact these results. We have added a comment noting this to the discussion.

7. Which time scale for the Cox Model?

Thank you for your question; we agree this needs to be further emphasized in our manuscript.

The time scale for the Cox Model is age in years, specifically age until death or lung transplant, whichever came first or age until censored for those that did not die or did not have lung transplant. In the Materials and Methods section Predictors and Outcome variables, this is specifically mentioned: “For the Cox model, time to event was set up as the minimum of the age of lung transplant or death, with age in the most recent review year used for censored observations.” We have added to this sentence to more thoroughly explain the time scale, added details indicating this in the Cox Multiple Regression Model section of the Results, and added to the Table 3 description. As noted in Fig. 2. Nomogram for probability of lung transplant or death free survival for 2 years, the measurement scale is in years. Although the nomogram is based on the Cox model results, it is depicting probability of lung transplant or death free survival for 2 years, defining time in years in the current moment to 2 years in the future.

Reviewer #2: This manuscript proposes a prediction model for lung transplantation and survival using standard logistic and Co

---

## [Decision Letter · Decision Letter 1]

13 May 2024

PONE-D-23-30531R1Modeling cystic fibrosis patient prognosis: nomograms to predict lung transplantation and survival prior to highly effective modular therapyPLOS ONE

Dear Dr. Piccorelli,

Thank you for submitting your manuscript to PLOS ONE. After careful consideration, we feel that it has merit but does not fully meet PLOS ONE’s publication criteria as it currently stands. Therefore, we invite you to submit a revised version of the manuscript that addresses the points raised during the review process.

**ACADEMIC EDITOR: Thanks for revising and resubmitting your work. Both reviewers have some additional comments to improve the current version and clarity for some of the findings. I think addressing following comments would improve your work.**==============================

We look forward to receiving your revised manuscript.

Kind regards,

Emrah Gecili, PhD

Academic Editor

PLOS ONE

Journal Requirements:

Reviewers' comments:

Reviewer's Responses to Questions

**Comments to the Author**

1. If the authors have adequately addressed your comments raised in a previous round of review and you feel that this manuscript is now acceptable for publication, you may indicate that here to bypass the “Comments to the Author” section, enter your conflict of interest statement in the “Confidential to Editor” section, and submit your "Accept" recommendation.

Reviewer #1: (No Response)

Reviewer #2: (No Response)

2. Is the manuscript technically sound, and do the data support the conclusions?

Reviewer #1: Yes

Reviewer #2: Partly

3. Has the statistical analysis been performed appropriately and rigorously? 

Reviewer #1: Yes

Reviewer #2: No

4. Have the authors made all data underlying the findings in their manuscript fully available?

Reviewer #1: No

Reviewer #2: Yes

5. Is the manuscript presented in an intelligible fashion and written in standard English?

Reviewer #1: Yes

Reviewer #2: Yes

6. Review Comments to the Author

Reviewer #1: Dear authors, thanks for taking into considerations my previous points.

Now I have only minor comments.

1.Patients who were already transplanted, were they excluded from the analysis? What is the prevalence of re-transplant in your dataset?

2. LTx as a competitive event. I would have expected an analysis taking this into consideration, e.g. multi-state modelling, as done by this group: 10.2147/TCRM.S391476. Btw, this is an European cohort mostly pre-HEMT which may bring additional discussion with regards to time-varying covariates.

3. 6 minute walking test. Actually this is a variable available in the CFF Registry: its role has been debated in CF for LAS calculation but its variation cound have been useful (doi:10.1080/09638288.2022.2099588), since the existent relationship between exercise capacity and survival.

4. I'd suggest to adapt the manuscript following these recommendations.

- Steyerberg EW, Moons KGM, van der Windt DA, et al. Prognosis Research Strategy (PROGRESS) 3: Prognostic

Model Research. Plos Medicine. 2013; 10.

- Collins GS, Dhiman P, Ma J, Schlussel MM, Archer L, Van Calster B, Harrell FE Jr, Martin GP, Moons KGM, van Smeden M, Sperrin M, Bullock GS, Riley RD. Evaluation of clinical prediction models (part 1): from development to external validation. BMJ. 2024 Jan 8;384:e074819. doi: 10.1136/bmj-2023-074819. PMID: 38191193; PMCID: PMC10772854.

- Collins GS, Reitsma JB, Altman DG, Moons KGM. Transparent reporting of a multivariable prediction model for individual prognosis or diagnosis (TRIPOD): the TRIPOD statement. BMJ 2015;350:g7594. doi:10.1136/bmj.g7594 18

- Moons KGM, Altman DG, Reitsma JB, et al. Transparent Reporting of a multivariable prediction model for Individual Prognosis or Diagnosis (TRIPOD): explanation and elaboration. Ann Intern Med 2015;162:W1-73. doi:10.7326/M14-0698

Reviewer #2: I still have clarity concerns as below.

1. For the correlations, authors can generate the correlation plot to include 20x20 correlation matrix (corplot in R). Given that 20 predictors are not immense, authors can create a secondary table (similar to Table 1) that summarizes each predictor missing frequency and refer to it instead of listing them in the main text.

2. Using AIC and p-value as guidance for inclusion of predictors will lead to overfitting given LARGE sample size. Authors need to compare the model with BIC or adjustedAIC were used for the model selection.

3. The rationale behind using complete case analysis is see whether there is any possibility of non-random missing data. Again, given this large sample size, imputation or not-impute data analysis can lead to the similar final model.

4. Authors can briefly discuss the goodness-of-fit test results, Hosmer-Lemeshow or RMS omnibus test results for logistic models and Schonfeld test results for Cox regression.

5. It is not clear how authors select the final model based upon this description "Full logistic and Cox regression models were fit including all independent variables of interest Significance of predictors (p ≤ 0.05),

Akaike information criterion (AIC), and concordance index were used to select the best subset of

predictors for the final models. The Holm method was used to correct p-values for multiple

testing [43]." Why authors considered multiple correction when multiple regression model was considered?

7. PLOS authors have the option to publish the peer review history of their article (what does this mean?). If published, this will include your full peer review and any attached files.

Reviewer #1: No

Reviewer #2: No

---

## [Author Response · Author response to Decision Letter 1]

25 Oct 2024

Reviewers' comments:

Reviewer's Responses to Questions

Comments to the Author

1. If the authors have adequately addressed your comments raised in a previous round of review and you feel that this manuscript is now acceptable for publication, you may indicate that here to bypass the “Comments to the Author” section, enter your conflict of interest statement in the “Confidential to Editor” section, and submit your "Accept" recommendation.

Reviewer #1: (No Response)

Reviewer #2: (No Response)

2. Is the manuscript technically sound, and do the data support the conclusions?

Reviewer #1: Yes

Reviewer #2: Partly

3. Has the statistical analysis been performed appropriately and rigorously? 

Reviewer #1: Yes

Reviewer #2: No

4. Have the authors made all data underlying the findings in their manuscript fully available?

Reviewer #1: No

Reviewer #2: Yes

5. Is the manuscript presented in an intelligible fashion and written in standard English?

Reviewer #1: Yes

Reviewer #2: Yes

6. Review Comments to the Author

Reviewer #1: Dear authors, thanks for taking into considerations my previous points.

Now I have only minor comments.

1.Patients who were already transplanted, were they excluded from the analysis? What is the prevalence of re-transplant in your dataset?

Age of transplant and the indicator of lung transplant was based on a variable that indicated the year of first lung transplant so patients’ lives post-lung transplant and patients with a re-transplant were not considered. We have an explanation of this to the manuscript.

2. LTx as a competitive event. I would have expected an analysis taking this into consideration, e.g. multi-state modelling, as done by this group: 10.2147/TCRM.S391476. Btw, this is an European cohort mostly pre-HEMT which may bring additional discussion with regards to time-varying covariates.

Thank you for bringing this manuscript to our attention. We agree that LTx is a competitive event and understand why Gambazza et al. (2023) used multistate modeling to address the effect of oxygen therapy on disease progression and we added a reference to this manuscript in our paper given its relevance in being a pre-HEMT study. We disagree that we should do multistate modeling with our study because the aim of our study is different. Gambazza et al. (2023) was looking at disease progression, whereas our study is predicting probability of lung transplant/death. We agree these outcomes are related to disease progression; however, we were trying to assess whether and when a patient should be listed for lung transplant. The way Gambazza et al. (2023) set up their multistate model with alive with and without lung transplant does not make sense in the context of our study. While it is possible for patients to require a second lung transplant we instead focused time to first lung transplant/death, whichever came first.

3. 6 minute walking test. Actually this is a variable available in the CFF Registry: its role has been debated in CF for LAS calculation but its variation cound have been useful (doi:10.1080/09638288.2022.2099588), since the existent relationship between exercise capacity and survival.

Thank you for bringing this manuscript to our attention. Given this reference about the 6-minute walking test was not published prior to when we requested the data in 2016, we were unaware of its significance for LAS calculation. We instead chose to request predictors that had been well-referenced prior to 2016, as noted in the manuscript. Given the potential for this variable for this to be useful, it was included in our description of potentially informative predictors in the limitations section in our discussion in the manuscript. We had previously included another reference supporting the use of this variable, so we added this reference to further support its use.

4. I'd suggest to adapt the manuscript following these recommendations.

- Steyerberg EW, Moons KGM, van der Windt DA, et al. Prognosis Research Strategy (PROGRESS) 3: Prognostic

Model Research. Plos Medicine. 2013; 10.

- Collins GS, Dhiman P, Ma J, Schlussel MM, Archer L, Van Calster B, Harrell FE Jr, Martin GP, Moons KGM, van Smeden M, Sperrin M, Bullock GS, Riley RD. Evaluation of clinical prediction models (part 1): from development to external validation. BMJ. 2024 Jan 8;384:e074819. doi: 10.1136/bmj-2023-074819. PMID: 38191193; PMCID: PMC10772854. 

Thank you for suggesting these manuscripts. We agree external validation and impact of using this model in clinical practice could provide further evidence of the benefits of using our models. We were unable to secure an external cohort in order to further validate our models and understand implementing them into clinical practice would require further study; however, we feel these models introduce a novel way of predicting probability and time to lung transplant that are more patient-specific than the current methods.

- Collins GS, Reitsma JB, Altman DG, Moons KGM. Tr3ansparent reporting of a multivariable prediction model for individual prognosis or diagnosis (TRIPOD): the TRIPOD statement. BMJ 2015;350:g7594. doi:10.1136/bmj.g7594 18

- Moons KGM, Altman DG, Reitsma JB, et al. Transparent Reporting of a multivariable prediction model for Individual Prognosis or Diagnosis (TRIPOD): explanation and elaboration. Ann Intern Med 2015;162:W1-73. doi:10.7326/M14-0698

Thank you for suggesting these manuscripts. We feel our manuscript thoroughly describes the items indicated as essential for reporting multivariable prediction models. 

Reviewer #2: I still have clarity concerns as below.

1. For the correlations, authors can generate the correlation plot to include 20x20 correlation matrix (corplot in R). Given that 20 predictors are not immense, authors can create a secondary table (similar to Table 1) that summarizes each predictor missing frequency and refer to it instead of listing them in the main text.

We agree it’s helpful to visualize the correlation among the predictors; however, given the variety of predictors in this study, we do not feel it’ll be helpful in this context. We have continuous variables (e.g. fev1pp, height), binary categorical (e.g. race, P. aeruginosa positive), and ordinal variables (e.g. CFRD and F508 genotype); thus, used a variety of different correlation measures to appropriately address the different combinations of each pair of variables. To look at the association between pairs of continuous variables, we used Pearson correlation, whereas, with pairs of binary variables, we used tetrachoric correlation, and with pairs of polychoric correlation—and we used these 3 types of correlation appropriately when the pairs of predictors were not of the same time. We feel providing a correlation plot with all these different types of correlation would be confusing and unnecessary given we didn’t set out to assess associations between predictors, but used the associations to inform the combination of predictors we used in our prediction models (recall—the prediction models were the primary aim of this study.). We feel our level of detail with regard to correlation is sufficient.

We agree mentioning the amount of missingness observed among the predictors is necessary but we don’t feel it is necessary to add a table to describe this. Not all predictors had missing observations so we feel setting up the predictors with missing observations in a table would highlight these more so than the predictors without missing observations.

2. Using AIC and p-value as guidance for inclusion of predictors will lead to overfitting given LARGE sample size. Authors need to compare the model with BIC or adjustedAIC were used for the model selection.

We agree adding BIC will help justify our selection predictors and we have added it to our methods and analysis. 

3. The rationale behind using complete case analysis is see whether there is any possibility of non-random missing data. Again, given this large sample size, imputation or not-impute data analysis can lead to the similar final model.

Thank you for your explanation of why you were interested in a complete case analysis. We agree that there is likely to be non-random missing data as we mentioned in our previous response: “In the case of our study population, it was possible that patients were lost to follow-up due to illness or being seen at a non-CF center hospital, so we did not feel comfortable assuming that the missing observations were not important.” We ran the complete case complete analysis and resulted in very similar results for both the logistic regression and the Cox regression models.

4. Authors can briefly discuss the goodness-of-fit test results, Hosmer-Lemeshow or RMS omnibus test results for logistic models and Schonfeld test results for Cox regression.

We agree a discussion of these will help more thoroughly describe our findings and have added a discussion of these to the manuscript. To assess the fit of the logistic model, we used a calibration plot of the predicted values against the observed values in order to visually observe and assess the fit, because the Hosmer-Lemeshow and RMS omnibus tests don’t allow that option.

5. It is not clear how authors select the final model based upon this description "Full logistic and Cox regression models were fit including all independent variables of interest Significance of predictors (p ≤ 0.05),

Akaike information criterion (AIC), and concordance index were used to select the best subset of

predictors for the final models. The Holm method was used to correct p-values for multiple

testing [43]." Why authors considered multiple correction when multiple regression model was considered?

Because the number of predictors in the final logistic and Cox models were 19 and 16, respectively, and given the sample size was large, we felt it was necessary to use the Holm method to minimize the potential for Type 1 error. With each step of model building, we checked the significance of the predictors, AIC, concordance index, and (upon this review) BIC, to determine if removing a predictor would improve the model, indicated by a meaningful decrease in AIC and BIC and a meaningful increase in concordance. The predictors were removed based on these and if their p-value was greater than 0.05. 

7. PLOS authors have the option to publish the peer review history of their article (what does this mean?). If published, this will include your full peer review and any attached files.

No.

Do you want your identity to be public for this peer review? For information about this choice, including consent withdrawal, please see our Privacy Policy.

Reviewer #1: No

Reviewer #2: No

---

## [Decision Letter · Decision Letter 2]

18 Nov 2024

Modeling cystic fibrosis patient prognosis: nomograms to predict lung transplantation and survival prior to highly effective modular therapy

PONE-D-23-30531R2

Dear Dr. Piccorelli,

We’re pleased to inform you that your manuscript has been judged scientifically suitable for publication and will be formally accepted for publication once it meets all outstanding technical requirements.

Kind regards,

Emrah Gecili, PhD

Academic Editor

PLOS ONE

Additional Editor Comments (optional):

Reviewers' comments:

Reviewer's Responses to Questions

**Comments to the Author**

1. If the authors have adequately addressed your comments raised in a previous round of review and you feel that this manuscript is now acceptable for publication, you may indicate that here to bypass the “Comments to the Author” section, enter your conflict of interest statement in the “Confidential to Editor” section, and submit your "Accept" recommendation.

Reviewer #1: All comments have been addressed

Reviewer #2: All comments have been addressed

2. Is the manuscript technically sound, and do the data support the conclusions?

Reviewer #1: Yes

Reviewer #2: Yes

3. Has the statistical analysis been performed appropriately and rigorously? 

Reviewer #1: Yes

Reviewer #2: Yes

4. Have the authors made all data underlying the findings in their manuscript fully available?

Reviewer #1: Yes

Reviewer #2: Yes

5. Is the manuscript presented in an intelligible fashion and written in standard English?

Reviewer #1: Yes

Reviewer #2: Yes

6. Review Comments to the Author

Reviewer #1: Dear authors, I think the clarity of the manuscript has generally much improved.

Thank you for the time spent reviewing the manuscript.

Reviewer #2: All major/minor concerns through a couple of rounds of revision were addressed adequately. I have no further comments.

7. PLOS authors have the option to publish the peer review history of their article (what does this mean?). If published, this will include your full peer review and any attached files.

Reviewer #1: No

Reviewer #2: No

---

## [Editor Report · Acceptance letter]

22 Nov 2024

PONE-D-23-30531R2 

PLOS ONE

Dear Dr. Piccorelli, 

I'm pleased to inform you that your manuscript has been deemed suitable for publication in PLOS ONE. Congratulations! Your manuscript is now being handed over to our production team.

Kind regards, 

on behalf of

Dr. Emrah Gecili 

Academic Editor

PLOS ONE